# Owl 🦉 : A Large Language Model for IT Operations

**Hongcheng Guo**[1], **Jian Yang**[1,*] **Jiaheng Liu**[1,*] **Liqun Yang**[1], **Linzheng Chai**[1],
**Jiaqi Bai**[1], **Junran Peng**[1], **Xiaorong Hu**[2], **Chao Chen**[2], **Dongfeng Zhang**[2],
**Xu Shi**[2], **Tieqiao Zheng**[2], **Liangfan Zheng**[2], **Bo Zhang**[2], **Ke Xu**[1], **Zhoujun Li**[1]
[1]State Key Lab of Software Development Environment, Beihang University
[2]Cloudwise Research
{hongchengguo,jiaya,liujiaheng,lqyang,bjq,lizj}@buaa.edu.cn
{tim.shi,steven.zheng,leven.zheng,bowen.zhang}@cloudwise.com

## Abstract

With the rapid advancement of IT operations, managing and analyzing large data volumes efficiently for practical applications has become increasingly critical. Natural Language Processing (NLP) techniques have demonstrated remarkable capabilities in various tasks, including named entity recognition, machine translation, and dialogue systems. Recently, Large Language Models (LLMs) have achieved significant improvements across various domain-specific areas. However, there is a noticeable gap in the development of specialized Large Language Models (LLMs) tailored for IT operations. In this paper, we introduce the Owl, a large language model trained on our constructed Owl-Instruct with a wide range of IT-related information. Specifically, limited by the maximum input length, we propose the **H**omogeneous **M**arkov **C**ontext **E**xtension method (HMCE). The mixture-of-adapter strategy is leveraged to improve the parameter-efficient tuning across different domains or tasks. Further, we evaluate the performance of Owl on the Owl-Bench established by us and open IT-related benchmarks. Owl demonstrates superior performance results on IT tasks, which outperforms existing models by significant margins. Moreover, we hope that the findings of our work will provide more insights to revolutionize the techniques of IT operations with specialized LLMs. Our code is available at https://github.com/HC-Guo/Owl.

## 1 Introduction

Large Language Models (LLMs) (Chowdhery et al., 2022b; Touvron et al., 2023a;b) have emerged as powerful tools in the field of natural language processing (NLP) and Artificial Intelligence (AI). The release of GPT-3 (Brown et al., 2020a) in 2020 demonstrated the advantages of training large auto-regressive LLMs. With 175 billion parameters, GPT-3 surpassed previous models in various LLM tasks including reading comprehension, question answering, and code generation. Similar results have been achieved by other models as well. Furthermore, evidence suggests that larger models exhibit emergent behaviors and possess abilities not present in smaller models. For instance, they can learn tasks from a few examples, a phenomenon known as few-shot prompting. This capability expands the scope of supported tasks and facilitates the automation of new language tasks for users. However, the majority of research endeavors have been directed towards constructing general Large Language Models (LLMs) that encompass a wide spectrum of subjects, certain models trained on domain-specific data have exhibited exceptional performance within their specific domains, such as science and medicine. These discoveries underscore the necessity for additional advancements in domain-specific models.

In the field of IT operations (Du et al., 2017; Liu et al., 2023; Guo et al., 2023a), the significance of natural language processing (NLP) technologies is steadily on the rise. This paper undertakes the crucial task of delineating a set of specific assignments within the realm of IT operations, encompassing areas such as information security, system architecture, and other domains. However,

---

[*]Corresponding authors.

the complexity and specific terminology of IT operations pose formidable challenges, including a unique set of terminologies, processes, and contextual nuances that are not easily decipherable by conventional NLP models. Therefore, it becomes increasingly evident that there is a pressing need for the development and deployment of a Large Language Model specifically tailored to the exigencies of IT operations within such specialized domains. The fine-tuned large language model customized to this purpose promises to be an invaluable asset in navigating the complexities of IT operations within these highly specialized domains. Such a specialized large language model would greatly enhance the efficiency, accuracy, and comprehension of IT-related tasks and communications within these niche areas, which will ultimately advance the field of IT operations management.

In this paper, we introduce the OWL, a large language model trained on our collected Owl-Instruct with a wide range of knowledge from operations and maintenance (O&M), where our data contains nine prevalent domains within O&M: information security, application, system architecture, software architecture, middleware, network, operating system, infrastructure, and database. Besides, we explore the use of the data augmentation (Xu et al., 2023b; Wang et al., 2023a) strategy to enable LLMs to accurately generate large, high-quality and diverse instruction data from a set of human-annotated data samples. To maintain a stringent standard of data quality, we employ a two-pronged approach that combines GPT-4 (OpenAI, 2023) scoring with meticulous manual validation. In addition, we have embarked on the development of Owl-Bench to evaluate the performance of different LLMs on IT-related tasks, an extensive bilingual benchmark that comprises two distinct segments: a Q&A (question-answer) part consisting of 317 entries, and a multiple-choice part containing 1,000 questions. In terms of model design, constrained by the maximum input length, inspired by the recent NBCE (Su, 2023) (Naive Bayes-based Context Extension), we propose the **H**omogeneous **M**arkov **C**ontext **E**xtension (HMCE) approach. Furthermore, in order to enhance the efficacy of instruction adaptation across various tasks, we put forward the strategy of employing a Mixture-of-Adapter method to facilitate supervised fine-tuning.

The contributions of our paper are as follows:

- Owl-Instruct Construction. We collect and label 3000 seed samples and then prompt ChatGPT (Ouyang et al., 2022) to generate diverse instructions. To cover practical scenarios, we curate instructions that involve both single-turn and multi-turn scenarios.
- Owl-Bench Construction. We have established a big model test benchmark for the operation and maintenance(O&M) domain to measure LLMs capabilities, which consists of nine O&M-related domains, showing the diversity of LLMs capabilities in the domain in a hierarchical manner.
- Model and Training. For tokenization, we expand the word vocabulary using the extra IT-related data. Besides, we propose a simple method to extend input length based on the homogeneous Markov chain. As for training, a Mixture-of-Adapter strategy is proposed to improve the instruction-tuning performance.
- Impressive Performance. We evaluate the performance of OWL with other LLMs on multiple benchmark datasets, including Owl-Bench and open IT-related benchmarks. OWL demonstrates superior performance results on IT tasks, which outperforms existing models by significant margins and maintains effective generalization abilities on Owl-Bench.

## 2 OWL-INSTRUCT DATASET CONSTRUCTION

The quality of the data employed in training Large Language Models (LLMs) stands as a pivotal determinant influencing the ultimate performance of the language model. Zhou et al. (2023) underscore the paramount importance attributed to both the diversity and quality of data when training large-scale models. Consequently, it becomes imperative for us to curate a high-caliber instruction dataset, which is the Owl-Instruct, tailored specifically for the realm of Operations and Maintenance (O&M). The overview of constructing Owl-Instruct is in Figure 1. A statistical analysis is presented in Table 1, and a visual representation of the keywords is depicted in Figure 2.

### 2.1 SEED DATA COLLECTION

In the initial phase of our project, we engage 25 subject matter experts within the field of operations and maintenance (O&M) to meticulously craft input and output sequences, along with comprehensive instructions. These encompass a wide spectrum of common domains and tasks. Concretely,

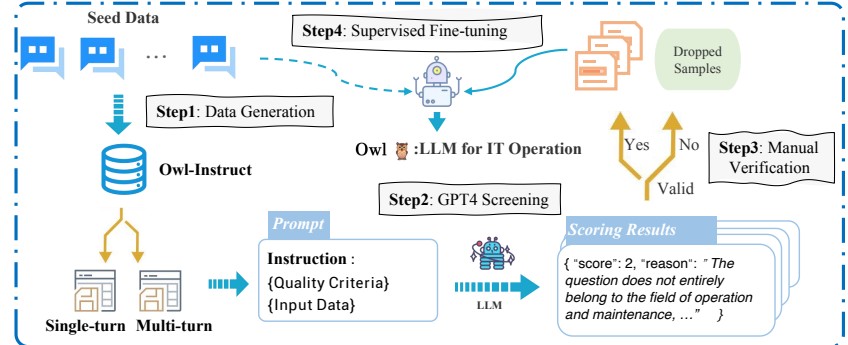

Figure 1: Four phases of constructing Owl-Instruct and how we train our OWL.

| Dataset | #Seed data | #Total dialogues | #Average turns | #Average dialogue length |
|---------|-----------|------------------|----------------|--------------------------|
| Single-turn | 2000 | 9118 | 1 | 335 |
| Multi-turn | 1000 | 8740 | 2.9 | 918 |

Table 1: Statistical analysis of the single-turn and multi-turn scenarios.

Owl-Instruct contains nine prevalent domains within O&M: information security, application, system architecture, software architecture, middleware, network, operating system, infrastructure, and database. Within each domain, a plethora of tasks are encapsulated, including but not limited to deployment, monitoring, fault diagnosis, performance optimization, log analysis, backup and recovery, among others. Finally, we gain 2,000 single-turn and 1,000 multi-turn seed data instances, which serve as the foundation for further augmenting the scale and diversity of Owl-Instruct.

## 2.2 DATASET AUGMENTATION

**Single-turn Dataset Construction** In our endeavor, we have constructed a comprehensive single-turn dialogue dataset tailored specifically to the domain of operations and maintenance, boasting an impressive 9,118 meticulously curated data entries. Motivated by Self-Instruct (Wang et al., 2023a), we have enriched our dataset. This enrichment involves the generation of supplementary samples derived from the seed data, a corpus painstakingly labeled by our domain experts. Besides, we consider GPT4 (OpenAI, 2023) as a reference and supervisor for ensuring the quality of data. Please refer to Appendix E.1 for cases.

**Multi-turn Dataset Construction** In accordance with the methodology elucidated in Baize (Xu et al., 2023b), the generation process of our multi-turn dialogue dataset within the operations and maintenance domain encompasses the following four distinct phases (Case in Appendix E.2.): (1) *Seed Data Collection*, (2) *Topic Generation*, (3) *Multi-turn Dialogue Generation*, and (4) *Manual and GPT4 Screening*. More details can be seen in Appendix C.

## 2.3 DATA QUALITY

In order to maintain a stringent standard of data quality, we employ a two-pronged approach that combines GPT-4 (OpenAI, 2023) scoring with meticulous manual validation. This dual-validation process ensures the integrity and reliability of generated data while enhancing its overall quality. When leveraging GPT-4 for scoring, we meticulously design specific prompts tailored to our dataset. These prompts are strategically crafted to enable GPT-4 to evaluate and rate the generated data based on predefined quality criteria. This automated scoring mechanism allows us to swiftly identify and filter out any low-quality data instances. Moreover, it serves as a valuable tool for flagging potential issues and areas that require improvement within the dataset. Simultaneously, dataset undergoes rigorous manual validation. A team of expert reviewers conducts an in-depth assessment of each data entry. This manual inspection process entails a thorough examination of content, coherence, and adherence to domain-specific knowledge. Entries that do not meet our stringent quality standards

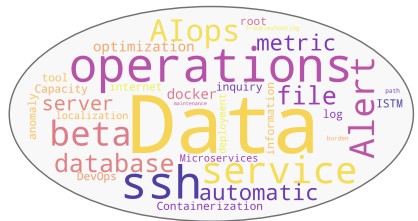

Figure 2: Keyword clouds in the Owl-Instruct. This keyword cloud is built based on word frequency. Words like "data", and "service" appear more frequently.

are meticulously flagged and subsequently removed. Please refer to Appendix F.1 for prompt and case details.

## 3 OWL-BENCH BENCHMARK CONSTRUCTION

**Overview:**  In the absence of a benchmark tailored for evaluating the performance of large language models in the context of Operations and Maintenance (O&M), there exists a critical gap in our ability to effectively assess and compare models within this area. To address this deficiency, we have embarked on the development of Owl-Bench, an extensive bilingual benchmark that comprises two distinct segments: a Q&A (question-answer) part consisting of 317 entries, and a multiple-choice part containing 1,000 questions in Table 2. The O&M is characterized by its vastness and multidisciplinary nature (Dang et al., 2019; Rijal et al., 2022; Gao & Pishdad-Bozorgi, 2019). Inspired by these works, we have meticulously curated the sub-fields included in Owl-Bench. We consider the multitude of real-world industrial scenarios that encompass this area, ensuring that our benchmark exhibits a comprehensive diversity. Our data collection process involves the acquisition of test data from nine distinct subdomains: information security, application, system architecture, software architecture, middleware, network, operating system, infrastructure, and database. And the data consists of Q&A pairs and multiple-choice questions. It is essential to highlight that the data incorporated into Owl-Bench significantly differs from that of Owl-Instruct. The former is derived directly from real-world scenario-based examination questions, devoid of any modification or expansion through GPT4. In Appendix E.3 and E.4, we present examples of multiple-choice questions and Q&A question across these nine domains, providing a glimpse into the rich diversity encapsulated within the benchmark.

### 3.1 DATA COLLECTION AND PROCESSING

Our main source of data consists of two parts: one part is a free practice exam available on the Internet. The other questions are carefully designed by O&M experts. More details can be seen in Appendix H. The data collected are in a variety of formats, mainly PDF or Microsoft Word documents, with a small percentage of web pages. The PDF documents are initially processed into text using OCR tools (Li et al., 2023). Some cases that are difficult to process will be manually parsed into a structured format by hand, similar to Hendrycks et al. (2021); Taylor et al. (2022).

| | Middleware | Information security | Infrastructure | Application | Operating system | Database | System architecture | Network | Software architecture |
|---|---|---|---|---|---|---|---|---|---|
| *Q&A* | | | | | | | | | |
| #Questions | 30 | 26 | 41 | 36 | 39 | 38 | 25 | 40 | 42 |
| #Average dialogue length | 301 | 275 | 287 | 311 | 297 | 342 | 286 | 308 | 298 |
| *Multiple-choice* | | | | | | | | | |
| #Questions | 136 | 108 | 110 | 102 | 118 | 119 | 87 | 122 | 98 |
| #Average dialogue length | 212 | 287 | 264 | 343 | 247 | 310 | 255 | 294 | 301 |

Table 2: Statistical analysis of the Owl-Bench.

## 4 MODEL

### 4.1 TOKENIZATION

Since LLaMA2 (Touvron et al., 2023b) is designed to support natural language in Latin or Cyrillic language families, it has unsatisfactory compatibility with IT operation data. For compatibility with

both natural language and IT-related data, we expand the word vocabulary size from $32,000$ to $48,553$. (More details in Appendix A)

## 4.2 LONG-CONTEXT INPUT

Limited by the maximum input length, inspired by the recent NBCE (Su, 2023) (Naive Bayes-based Context Extension), we propose the **H**omogeneous **M**arkov **C**ontext **E**xtension method (HMCE). The fundamental assumption of NBCE is the independent input contexts, but in reality, input texts are interconnected and exhibit continuity rather than independence. Consequently, we abandon the independence assumption and instead employ a homogeneous Markov chain assumption to extend NBCE, catering to the continuous nature of the input data, which is named HMCE.

Let $T$ be the generated text. Let $S := (S_1, ..., S_n)$ be the previous contexts, and that their overall length has exceeded the training length. We want to generate $T$ based on $S_1, ..., S_n$, that is, we want to estimate $p(T|S)$. (See Appendix D.2 for more details and derivation.)

$$\log p(T|S) \propto \sum_{i=2}^{n} p(T|S_i, S_{i-1}) - \sum_{i=2}^{n-1} p(T|S_i) \tag{1}$$

where $n$ is the number of separate content. For the original input sentences $S$ longer than maximum length $L$, we slit them into multiple segments.

## 4.3 MIXTURE-OF-ADAPTER

Parameter-efficient tuning (Houlsby et al., 2019; Zhu et al., 2021; Hu et al., 2022; Yang et al., 2022; He et al., 2022; Wang et al., 2023b; 2022) is a simple but effective technique in LLMs, which enables efficient and flexible transfer by introducing task-specific modifications to a fixed pre-trained model. For the cross-domain/cross-task transfer, we use a mixture of adapters for different domains and tasks, where a group of LoRA adapters is lightweight compared to the pre-trained model. The adapters with a low-rank down-project matrix and up-project matrix can be directly inserted into the pre-trained embedding, attention, and feed-forward network. Given the source sentence $x = \{x_1, \ldots, x_m\}$ of $m$ tokens and a group of $T$ Adapters, we use mixture-of-adapters to learn the task-specific and domain-specific representations for diverse input:

$$h_a^{L_i} = \mathcal{A}_{\theta_{g(L_i)}}(h^{L_i}) \tag{2}$$

where $g(L_i)$ are selected LoRA experts derived from the language representations. $\mathcal{A}(\cdot)$ denotes the LoRA adapter module and $\theta = \{\theta_1, \ldots, \theta_T\}$ denotes the adapter pool. $\mathcal{A}_{\theta_{g(L_i)}}$ is calculated by:

$$\mathcal{A}_{\theta_{g(L_i)}}(h^{L_i}) = h^{L_i} + \sum_{A_t, B_t \in \mathcal{S}(e)} \alpha \Delta W h^{L_i} \tag{3}$$

where $\Delta W = BA$ is denoted by a low-rank decomposition ($A \in \mathbb{R}^{d \times r} \wedge B \in \mathbb{R}^{r \times d} \wedge r \ll d$). The matrices $A$ and $B$ are initialized by a random Gaussian distribution and zero. $\alpha$ is the scaling factor and $r$ is the inner dimension. $\mathcal{S}(e)$ denotes the subset.

In Equation 3, all experts only require fine-tuning a small number of language-specific parameters instead of all parameters of the pre-trained model. Thus, we can simultaneously train multiple experts for different languages, which all share the same freezing pre-trained parameters. We use multiple adapters from the selected subset to maximize the transfer of knowledge across languages:

$$g(L_i) = \texttt{TopK}\left(\frac{exp(\alpha_j^{L_i})}{\sum_{t=1}^{T} exp(\alpha_t^{L_i})}\right) \tag{4}$$

where $\texttt{TopK}(\cdot)$ is the selection function, where we calculate the selection probabilities of all LoRA adapters and choose the top-$k$ LoRA experts obeying the probability distribution. $\alpha_j^{L_i}$ is a scalar from the representations of language $L_i$ (We use the hidden state of the special token [CLS] of each layer). $\mathcal{S}(e) = \{(A_k, B_k)\}_{k=1}^{K}$ and $\alpha_j^{L_i}$ is used to incorporate the different experts.

We project the language representation $e^{L_i}$ of language $L_i$ into the LoRA expert distribution using the learned matrix $W_a \in \mathbb{R}^{d \times T}$, where $d$ is the hidden size and $T$ is the number of experts. The weight of LoRA expert $\alpha_j^{L_i}$ is calculated by:

$$\alpha^{L_i} = e^{L_i} W_a \tag{5}$$

where $\alpha = \{\alpha_t = 1\}_{t=1}^T$. For all modules of the pre-trained model, we leverage the mixture-of-adapter strategy to learn the task-sensitive representations for the different input sentences by activating top-$k$ experts.

## 4.4 SUPERVISED FINE-TUNING

Given the multiple tasks $T = \{T_i\}_{i=1}^N$, we construct the multi-task training corpora $D = \{D_i\}_{i=1}^N$, where each dataset contains a series of triple tuples $\{(x^{(j)}, y^{(j)}, I^{(j)})\}_{j=1}^M$, where $x^{(j)}$ and $y^{(j)}$ are input and output sample with the instruction $I^{(j)}$.

**Instruction Tuning at Scale**  To scale up the multi-task training corpora $D = \{D_i\}_{i=1}^N$, we adopt the self-instruction (Wang et al., 2023a) for increasing data diversity and complexity. Given the seed human-written instructions as in-context examples (randomly sampling $K$ task instruction from the initial instruction pool), new instructions are generated and merged into the instruction pool. The process is repeated several times until no new legitimate instructions are generated. Based on task descriptions and their instructions, we expand the training data of each instruction by leveraging a large language model to output the sample input and then produce the answer. To filter out the low-quality data, we score the model-generated samples by feeding the generated sample into the LLM, and three human experts fix or drop out the illegal samples (the scoring prompt is shown in Section F.1). After the heuristic filter process, a new high-scoring instruction passing expert check will be added to the instruction pool. The model-generated training corpora $D^m = \{D_i^m\}_{i=1}^N$ are merged into original training corpora as a whole $D \bigcup D^m$ for multi-task training.

**Multi-task Training**  Given the supervised and model-generated instruction corpora $D \bigcup D^m$, the training objective of the supervised instruction tuning can be described as:

$$\mathcal{L}_m = -\frac{1}{N} \sum_{i=1}^N \mathbb{E}_{x,y,I \in \{D_i, D_i^m\}} \log(y^{(i)} | I^{(i)}, x^{(i)}) \tag{6}$$

where $x$ is the sample input and $y$ is the sample output with the instruction $I$ from the original training corpora and model-generated training corpora.

## 5 EVALUATION

We evaluate the performance of OWL on Owl-Bench and general downstream tasks, where the Owl-Bench help us test our hypothesis that training on high-quality data will yield better results on the questions in operation and maintenance area. The general tasks investigate whether OWL has the generalization capability. For general downstream tasks, we chose two typical benchmarks: Log Parsing and Anomaly Detection Tasks.

## 5.1 EXPERIMENT SETTING

Our base model is LLaMA2-13b (Touvron et al., 2023b). For instruction-tuning, the learning rate is $10^{-4}$, a weight decay of 0.1, a batch size of 16. The sequence length is 1024. We use Adam as the optimization algorithm with $\beta_1 = 0.9$, $\beta_2 = 0.99$, and $\varepsilon = 10^{-8}$. The training epoch is 3. The rank and alpha of LoRA (Hu et al., 2022) is 8 and 32. The dropout of LoRA is 0.05. We train LoRA for 10 epochs.

## 5.2 EVALUATION ON OWL-BENCH

In this section, we compare the results of most of the large language models (LLMs) on our benchmark (Owl-Bench). The results of the experiment consist of two main parts: the results of the

|  | LLaMA2-13b | ChatGLM-6b | ChatGLM2-6b | Qwen-7b | InternLM-7b | OWL-13b |
|---|---|---|---|---|---|---|
| Average score | 8.57 | 8.12 | 8.27 | 8.41 | 8.19 | **8.86** |

Table 3: Average scores on the Q&A part of Owl-Bench. Scores range from 1 to 10.

multiple choice questions and the Q&A test. The multiple choice questions mainly test the model's objective domain general knowledge learning ability, and the Q&A questions mainly test the model's comprehensive processing and logic ability for O&M problems. The models we choose to compare have similar sizes to Owl and are open-sourced that the results can reproduced: ChatGLM2-6b (Du et al., 2022), ChaGLM-6b (Du et al., 2022), LLaMA2-13b (Touvron et al., 2023b), Qwen-7b [1] and InternLM-7b (Team, 2023). Besides, we also compare our model with ChatGPT (Jiao et al., 2023).

### 5.2.1 RESULTS ON Q&A TEST

**Evaluation way**    Follow recent works (Huang et al., 2023; Xu et al., 2023b), there are two ways for evaluation: single-score mode and pairwise-score mode. For single-score mode, firstly, we select the model which will be tested, and let the model give the answer directly based on the given questions. Then, we choose the scoring model (GPT4 (OpenAI, 2023)), and let the scoring model give a score from 1 to 10 based on the question content. Higher scoring values indicate better responses. Please refer to the Appendix F.2 for more details.

For the pairwise-score mode, inspired by the work (Zheng et al., 2023), first, we select two models which will be assessed, and let the models give the answers based on the given questions. Then, let the scoring model judge which model among the assessed models is better according to the question content, the answers of the two assessed models, and the reference answer. The better model counts one win, the worse model counts one loss, and if the models answer at a similar level, they all count one tie. Please refer to the Appendix F.3 for more details.

**Q&A Performance**    In Table 3, we show the average scores of the Q&A test, where OWL gets the highest score, and overall the models perform well, with a small variance in the scores. As for the pairwise scores, we can see in Figure 3 that OWL also beats the rest of the models with the best performance, but the number of draws is on the high side, which means in most cases, both generate answers to the satisfaction of the GPT4.

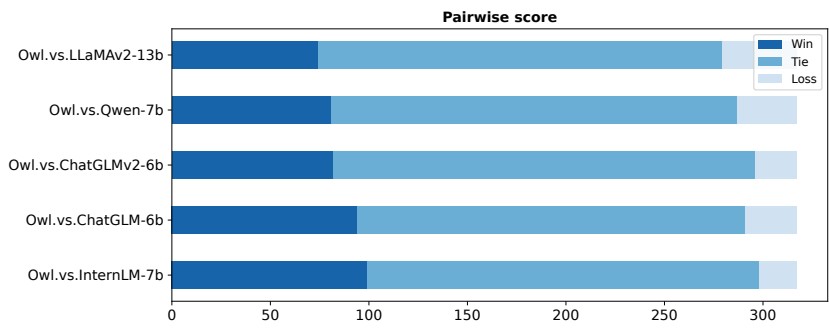

Figure 3: Pairwise scores of different models on Q&A test in Owl-Bench.

### 5.2.2 RESULTS ON MULTIPLE CHOICE QUESTIONS

**Evaluation way**    For the multiple-choice questions, our approach involves direct model response generation by allowing the model to select a single answer from the provided options (A, B, C, D). This way not only streamlines the evaluation process but also offers a standardized format for assessing the model's comprehension and decision-making abilities. Each question presents a set of choices, and the model's task is to make a single selection that best aligns with its understanding

---
[1]https://github.com/QwenLM/Qwen-7B

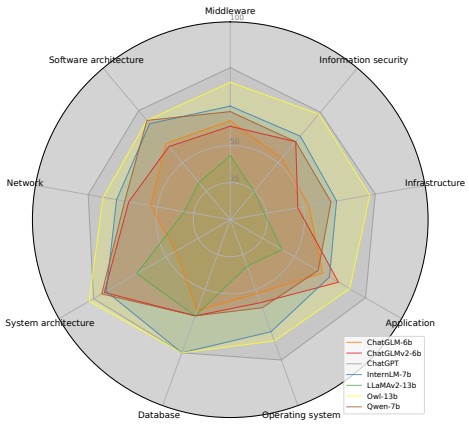 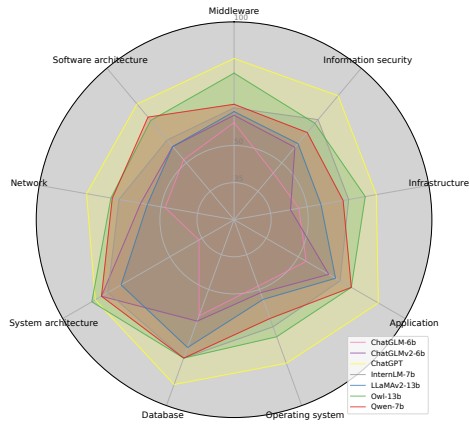

(a) Zero-shot results on different models.

(b) Five-shot results on different models.

Figure 4: Zero-shot/Five-shot results of multiple-choice questions on Owl-Bench.

of the question's context and semantics. This assessment approach is widely employed in educational and evaluative contexts to gauge a model's accuracy and proficiency in selecting the correct answer among multiple possibilities. It enables a straightforward and efficient means of evaluating the model's performance in a multiple-choice question setting, facilitating a clear and concise assessment process.

**Multiple choice performance** The results in Figure 4a and Figure 4b show that ChatGPT (Jiao et al., 2023) has the highest average scores among these 9 domains, and OWL is also just a little bit lower than ChatGPT, while outperforming the other models. Besides, OWL achieves a higher point than ChatGPT on the system architecture domain.

## 5.3 EVALUATION ON LONG CONTEXT INPUT

| Model | Sequence Length | | | |
|---|---|---|---|---|
| | 1024 | 2048 | 4096 | 8192 |
| OWL-13b | 4.34 | 22.79 | 106.33 | 314.49 |
| OWL-13b + NBCE (Su, 2023) | 4.34 | 6.18 | 7.35 | 8.24 |
| OWL-13b + HMCE | 4.34 | 5.02 | 5.57 | 6.10 |

Table 4: Results of OWL on long-context inference using various techniques.

We have propose HMCE for training-free long-context inference based on NBCE (Su, 2023). Specifically, HMCE employs a conditional homogeneous Markov chain to extend the context processing capabilities of large language models (LLMs). Notably, this extension is model-agnostic and necessitates no fine-tuning. To substantiate the efficacy of this approach, we randomly selected Q&A pairs from the Owl-Bench and concatenated the questions to create input sequences of varying lengths. The results in Table 4 are assessed in terms of perplexity (PPL). These findings unequivocally underscore the effectiveness of HMCE. Without NMCE, the PPL experiences a proportional increase as the input length grows.

## 5.4 EFFECT OF MIXTURE-OF-ADAPTER

In Table 5, we perform ablation experiments to compare the results of multiple choice with and without Mixture-of-Adapter (MoA) strategy, and the experiments show that our MoA is effective and we also compared it with the fashion LoRA (Hu et al., 2022), which better shows the efficiency of our model. Specifically, when using instruction-tuning without MoA, the overall performance of

the model is slightly degraded, and when using LoRA fine-tuning, the final performance is close to the result without MoA.

| Method | Middleware | Information security | Infrastructure | Application | Operating system | Database | System architecture | Network | Software architecture | Mean |
|---|---|---|---|---|---|---|---|---|---|---|
| | | | | *Zero-shot Testing Performance* | | | | | | |
| Owl w/o MoA | 0.70 | 0.72 | 0.74 | 0.73 | 0.69 | 0.75 | 0.85 | 0.73 | 0.70 | 0.73 |
| Owl w/ LoRA | 0.69 | 0.70 | 0.72 | 0.70 | 0.65 | 0.71 | 0.82 | 0.71 | 0.72 | 0.71 |
| Owl w/ MoA | 0.75 | 0.76 | 0.77 | 0.75 | 0.72 | 0.77 | 0.86 | 0.72 | 0.72 | 0.76 |

Table 5: Effect of Mixture-of-Adapter (MoA). For the Owl without MoA, we directly perform instruct-tuning on the base model. Here, the Zero-shot testing accuracies are provided.

## 5.5 EVALUATION ON DOWNSTREAM TASK

### 5.5.1 LOG ANOMALY DETECTION

**Task Description** Log anomaly detection represents a crucial component of automated log analytics, employed for real-time system issue detection for large-scale IT systems. Anomaly detection, as elucidated in the work (Breier & Branišová, 2015), assumes paramount importance in scrutinizing idiosyncrasies within log data. These logs provide intricate insights into system events transpiring in real-time, as well as user intentions within the ambit of large-scale services, as articulated in the study (Zhang et al., 2015). The endeavor to pinpoint anomalous logs solely from a local perspective is fraught with potential errors.

**Datasets and Baselines** We conduct experiments on LogHub (He et al., 2020). Baseline methods: DeepLog (Du et al., 2017), LogAnomaly (Meng et al., 2019), LogRobust (Zhang et al., 2019), and LogPrompt (Liu et al., 2023). The basic settings are consistent with LogPrompt to ensure the zero-shot scenario. Specifically, we use the first 4000 log messages in each dataset to train and then test on the remaining logs. The evaluation metric is the F1-score, wherein $TP$ signifies the successful detection of an anomalous session (likewise for $TN$, $FP$, and $FN$). The prompts utilized for ChatGPT and OWL are in Appendix F.5:

| Methods | BGL | | | | Spirit | | | |
|---|---|---|---|---|---|---|---|---|
| | T.N.[a] | P | R | F | T.N. | P | R | F |
| DeepLog (Du et al., 2017) | 4000 | 0.156 | **0.939** | 0.268 | 4000 | 0.249 | 0.289 | 0.267 |
| LogAnomaly (Meng et al., 2019) | 4000 | 0.016 | 0.056 | 0.025 | 4000 | 0.231 | 0.141 | 0.175 |
| LogRobust (Zhang et al., 2019) | 4000 | 0.095 | 0.425 | 0.156 | 4000 | 0.109 | 0.135 | 0.120 |
| LogPrompt(ChatGPT) (Liu et al., 2023) | 0 | 0.249 | 0.834 | 0.384 | 0 | 0.290 | **0.999** | 0.450 |
| **Owl** | 0 | **0.301** | 0.866 | **0.446** | **0** | **0.354** | 0.972 | **0.518** |

[a] **T.N.** denotes the number of logs utilized for training.
  **P** denotes Precision. **R** denotes Recall. **F1** denotes F1-score.

Table 6: Log Anomaly Detection in the Zero-shot Scenario.

**Results and Analysis** The results are depicted in Table 6. Remarkably, OWL outperforms existing methods in both datasets, despite the latter being trained on thousands of logs. In a comparison with ChatGPT (LogPrompt), OWL exhibits an average improvement of 6.5% in terms of F1-score across the two datasets. Nevertheless, even with the formidable Large Language Model capabilities, the performance of anomaly detection in the zero-shot scenario remains modest.

Model results on the Log Parsing downstream task are shown in Appendix B.

## 6 CONCLUSION

In this paper, we present the OWL, a large language model for IT operations. First, we collect the Owl-Instruct dataset, which contains diverse IT-related tasks to improve the generalization ability of LLMs on IT operations. Then, we also introduce the Owl-Bench evaluation benchmark dataset with nine operation and maintenance domains. Besides, we also introduce the HMCE method to extend the context and the mixture-of-adapter strategy to further enhance the performance. Moreover, extensive experimental results on Owl-Bench demonstrate the effectiveness of our OWL.

## 7 ACKNOWLEDGEMENT

This work was supported in part by the National Natural Science Foundation of China (Grant Nos. 62276017, U1636211, 61672081), and the Fund of the State Key Laboratory of Software Development Environment (Grant No. SKLSDE-2021ZX-18).

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

| Methods | T.R.[a] | HDFS | | Hadoop | | Zookeeper | | BGL | | HPC | | Linux | | Proxifier | | Android | | Avg. | |
|---|---|---|---|---|---|---|---|---|---|---|---|---|---|---|---|---|---|---|---|
| | | RI[b] | F1 | RI | F1 | RI | F1 | RI | F1 | RI | F1 | RI | F1 | RI | F1 | RI | F1 | RI | F1 |
| IPLoM Makanju et al. (2009) | | 0.914 | 0.389 | 0.636 | 0.068 | 0.787 | 0.225 | 0.858 | 0.391 | 0.228 | 0.002 | 0.695 | 0.225 | **0.822** | 0.500 | 0.918 | 0.419 | 0.733 | 0.277 |
| LKE Fu et al. (2009) | | 0.861 | 0.424 | 0.150 | 0.198 | 0.787 | 0.225 | 0.848 | 0.379 | 0.119 | 0.381 | 0.825 | 0.388 | 0.379 | 0.309 | 0.045 | 0.000 | 0.502 | 0.288 |
| LogSig Tang et al. (2011) | | 0.872 | 0.344 | 0.651 | 0.050 | 0.787 | 0.225 | 0.806 | 0.333 | 0.119 | 0.002 | 0.715 | 0.146 | 0.559 | 0.339 | 0.732 | 0.116 | 0.655 | 0.194 |
| FT-tree Zhang et al. (2017) | 10% | 0.908 | 0.385 | 0.668 | 0.046 | 0.773 | 0.186 | 0.275 | 0.497 | 0.119 | 0.002 | 0.709 | 0.211 | 0.722 | 0.420 | 0.918 | 0.581 | 0.636 | 0.291 |
| Spell Du & Li (2016) | | 0.871 | 0.000 | 0.721 | 0.058 | 0.102 | 0.045 | 0.503 | 0.536 | 0.882 | 0.000 | 0.706 | 0.091 | 0.621 | 0.000 | 0.822 | 0.245 | 0.654 | 0.122 |
| Drain He et al. (2017) | | 0.914 | 0.389 | 0.647 | 0.068 | 0.787 | 0.225 | 0.822 | 0.397 | 0.119 | 0.002 | 0.695 | 0.225 | **0.822** | 0.500 | 0.916 | 0.413 | 0.716 | 0.277 |
| MoLFI Messaoudi et al. (2018) | | 0.871 | 0.000 | 0.699 | 0.095 | 0.899 | 0.000 | 0.792 | 0.333 | 0.881 | 0.000 | 0.410 | 0.026 | 0.621 | 0.000 | 0.173 | 0.208 | 0.668 | 0.083 |
| LogParse Meng et al. (2020) | | 0.907 | 0.632 | 0.349 | 0.502 | 0.982 | 0.348 | **0.992** | 0.665 | 0.194 | 0.330 | 0.825 | 0.588 | 0.490 | 0.334 | 0.288 | 0.233 | 0.628 | 0.454 |
| LogStamp Tao et al. (2022) | | 0.954 | 0.523 | 0.927 | 0.594 | **0.992** | 0.275 | 0.984 | 0.818 | **0.949** | 0.434 | 0.760 | 0.658 | 0.811 | 0.438 | 0.974 | **0.899** | **0.919** | 0.580 |
| LogPPT Le & Zhang (2023) | 0.05% | **0.960** | 0.838 | **0.987** | 0.526 | 0.988 | 0.795 | 0.859 | **0.982** | 0.238 | 0.287 | **0.831** | 0.423 | 0.804 | 0.638 | 0.782 | 0.313 | 0.806 | 0.600 |
| LogPrompt(ChatGPT) Liu et al. (2023) | 0% | 0.890 | 0.927 | 0.879 | 0.862 | 0.948 | 0.934 | 0.964 | 0.943 | 0.934 | 0.796 | 0.758 | **0.860** | 0.567 | **0.998** | **0.978** | 0.725 | 0.865 | 0.881 |
| **Owl** | 0% | 0.916 | **0.933** | 0.925 | **0.898** | 0.966 | **0.955** | 0.941 | 0.932 | 0.946 | **0.812** | 0.737 | 0.849 | 0.704 | 0.968 | 0.959 | 0.804 | 0.886 | **0.894** |

[a] **T.R.** denotes training ratio, the ratio of logs utilized for training.
[b] **RI** stands for RandIndex. **F1** stands for fine-level F1-score.

Table 7: Performances of Log Parsing in the Zero-shot Scenario.

## A TOKENIZATION

Since LLaMA Touvron et al. (2023a) is designed to support natural language in Latin or Cyrillic language families, it has unsatisfactory compatibility with IT operation data (The vocabulary of LLaMA only contains 32K words). When tokenization is performed on IT operation data, a terminology of IT operation is often split into multiple parts (2-3 tokens are required to combine a term of log data), which significantly reduces the efficiency of coding and decoding. For compatibility with both natural language and log data, we expand the word vocabulary using the extra data. Specifically, we train a tokenizer model on Owl-Instruct dataset and then merge the LLama tokenizer with the LLaMA native tokenizer by combining their vocabularies for a coupled tokenizer model. Consequently, we obtain a merged tokenizer with a vocabulary size of 48,553. To adapt the LLaMA model for the new tokenizer, we resize the word embeddings and language model head from shape $D \times T$ to $D' \times T$, where $D = 32,000$ denotes the original vocabulary size, and $D' = 48,553$ is the new vocabulary size. The new rows are appended to the end of the original embedding matrices, ensuring that the embeddings of the tokens in the original vocabulary remain unaffected.

## B RESULTS ON LOG PARSING TASK

**Task description** Log parsing represents a classical challenge within the realm of log analysis. Although existing approaches have made noteworthy strides in log analysis, they encounter substantial challenges when deployed in real-world scenarios. Firstly, the performance of current methods experiences a marked decline when confronted with situations characterized by a scarcity of training samples, coupled with a preponderance of previously unseen logs.

Secondly, the pragmatic implementation of log analysis techniques is hampered by their limited interpretability. Conventional methods furnish predictions devoid of accompanying explanations. Conversely, an interpretable output from the analysis not only facilitates the identification of false alarms but also simplifies the task of tracing the root causes of issues and subsequently taking appropriate corrective measures.

**Datasets and Baselines** We have conducted experiments on LogHub benchmark He et al. (2020). To assess the log parsing performance under zero-shot conditions, we adopt the same setting outlined in LogPrompt Liu et al. (2023). 10 baselines are chosen, including LKE Fu et al. (2009), LogSig Tang et al. (2011), Spell Du & Li (2016), IPLoM Makanju et al. (2009), Drain He et al. (2017), FT-tree Zhang et al. (2017), MoLFI Messaoudi et al. (2018), Logstamp Tao et al. (2022), LogPPT Le & Zhang (2023), and LogPrompt Liu et al. (2023).

Evaluation employs the same metrics as LogPrompt: RandIndex Rand (1971); Tao et al. (2022); Zhang et al. (2017); Meng et al. (2020) and fine-level F1-score. To adhere to the zero-shot paradigm, we follow the LogPrompt: for each dataset, the baselines are trained on the initial 10% of the logs and then evaluated on the remaining 90%. Specifically, LogPPT is trained on only the first 0.05% of the logs, while LogPrompt and Owl are directly tested on the remaining 90% of the logs. The prompt for parsing is in Appendix F.4.

**Results and Analysis** The results are shown in Table 7. Remarkably, despite its lack of training data, Owl achieves comparable performance on the RandIndex and the best F1 scores. For RandIndex comparison, Owl exhibits only marginal performance degradation than LogStamp. In the realm of fine-level F1 comparisons, Owl outperforms other baselines significantly, displaying a remarkable capacity to accurately identify variables within previously unseen logs. Notably, the foundational model for logPrompt is ChatGPT Ouyang et al. (2022). When compared to ChatGPT under identical fundamental settings, Owl delivers superior performance, underscoring the robust generalization capabilities in operations and maintenance (O&M).

## C   MORE DETAILS ON MULTI-TURN DATASET CONSTRUCTION

- **Seed Data Collection**: This initial phase involves the meticulous curation of original seed data, a collection painstakingly annotated by domain experts renowned within the operations and maintenance field.

- **Topic Generation**: Building upon the seed data acquired in the preceding step, we harness the GPT-4 OpenAI (2023) to generate a myriad of topics. This process is meticulously designed to ensure that the generated content remains firmly rooted within the operations and maintenance domain, all while maintaining a desirable level of topic diversity.

- **Multi-turn Dialogue Generation**: In this critical stage, we employ the Baize multi-turn dialogue generation method Xu et al. (2023b). Leveraging the topics forged in the prior phase, this method artfully crafts multi-turn dialogue data that is intrinsically tied to the operations and maintenance domain.

- **Manual and GPT4 Screening**: To further bolster the quality of our generated data, we enlist the capabilities of GPT-4 OpenAI (2023). In addition to the automated screening, our dataset undergoes rigorous manual inspection and cross-validation. This meticulous process ensures that each data entry is meticulously reviewed by at least three individuals. Entries that fail to meet the high-quality standards are promptly removed. Consequently, our comprehensive multi-turn dataset comprises a total of 8,740 dialogue entries, with an average of approximately three turns per dialogue.

## D   MORE DETAILS ON MODEL

### D.1   ROTARY EMBEDDING

Language models based on Transformers use a self-attention mechanism to consider the positional information of individual tokens, which facilitates the exchange of knowledge between tokens located at various positions. The multi-head attention can be described as:

$$X_{attn} = \mathop{\big\|}_{h=1}^{H} \text{SF} \left( \frac{QK^T}{\sqrt{d_k}} \right) V \tag{7}$$

where $\text{SF}(\cdot)$ is the softmax function, and $\|_{h=1}^{H}$ is the feature concatenation of the $H$ attention heads. The input representation is projected into $Q, K, V$ with the learned matrix.

To account for relative position information, we need a function $g$ that operates on word embeddings $x_m, x_n$ and their relative position $(m - n)$ as input variables. The objective is to ensure that the inner product of query $q_m$ and key $k_n$ encode position information solely in its relative form. To incorporate position information with self-attention, we inject the position information into the query and key, respectively. We set the inner product of these two terms to a function explicitly depending on their relative distance as:

$$f_{\{q,k\}}(x_m, m) = R^d_{\Theta,m} W_{\{q,k\}} x_m \tag{8}$$

where $R^d_{\Theta,m}$ is the rotary matrix with pre-defined parameters for incorporating the relative position information into the attention mechanism.

## D.2 DETAILS ON HMCE

Let $T$ be the text that needs to be generated. Let $S := (S_1, ..., S_n)$ be the previous contexts, and that their overall length has exceeded the training length. We want to generate $T$ based on $S_1, ..., S_n$, that is, we want to estimate $p(T|S)$.

We assume that $S$ follows a conditional homogeneous Markov chain structure, that is

$$p(S_i|S_{1:i-1}, T) = p(S_i|S_{i-1}, T) \tag{9}$$

Also, by the Bayes' formula and the conditional probability, we have

$$p(T|S) \propto p(S|T)p(T) \tag{10}$$

$$p(S, T) = p(S|T)p(T) = p(S_n|S_{1:n-1}, T)p(S_{1:n-1}, T) \tag{11}$$

Thus we gain the final log-likelihood of $P(T|S)$. The detailed derivation is in Appendix G.2.

$$\log p(T|S) \propto \sum_{i=2}^{n} p(T|S_i, S_{i-1}) - \sum_{i=2}^{n-1} p(T|S_i) \tag{12}$$

where $n$ is the number of separate content. For the original input sentences $S$ longer than maximum length $L$, we slit them into multiple segments and feed them into the model to compute their corresponding log-likelihoods and finally obtain the log-likelihood of the original input sentence. Besides, we compare Eq. 14 and Eq. 12 and find that NBCE is a special case of HMCE.

## E OWL-INSTRUCT AND OWL-BENCH EXAMPLES

Here we give some specific examples from the Owl-Instruct and Owl-Bench.

### E.1 SINGLE-TURN CASE IN OWL-INSTRUCT

> **Single-turn Example Structure:**
> {"q": "Please answer the following questions: The server is running slowly and how to troubleshoot and solve it",
> "a": "1. Check whether the server hardware resources are overloaded, such as CPU, memory, disk space, etc. 2. Check whether the applications running on the server are running normally, such as checking application logs, monitoring alarms, etc." }

### E.2 MULTI-TURN CASE IN OWL-INSTRUCT

> **Multi-turn Example Structure:**
> {"id": "0", "topic": "How to monitor and manage logs in distributed systems?", "conversations": [{"[question]": "I have a question about distributed systems. Can you tell me how to monitor and manage logs in distributed systems?", "[answer]": "When monitoring and managing logs in distributed systems, the following are some common practices and tools: 1. Central log collector: Use a central log collector such as Elasticsearch, Logstash, and Kibana (ELK) or Splunk and other tools..."}, {"[question]": "Among the tools you mentioned, I am familiar with Elasticsearch and Kibana. Can you explain in detail how to use these two tools to monitor and manage logs in distributed systems?", "[answer]": "When using Elasticsearch and Kibana to monitor and manage logs in a distributed system, you can follow the following steps:1. Install and configure Elasticsearch: First, install and configure Elasticsearch as the central log storage and search engines..."}, {"[question]": "...", "[answer]": "..."}] }

## E.3 Q&A QUESTION IN OWL-BENCH

> **Case of Q&A question in Owl-Bench**
> ##Question:
> {"id": 1, "domain": "Information Security", "question": ["How to protect system security?"]} ##Reference
> {"id": 1, "answer": ["The following are several key steps to protect system security: 1. Install security updates: Ensure that all security updates for the system's operating system, applications, and anti-virus software have been installed , to ensure that the system is not vulnerable to attacks due to known vulnerabilities. 2. Use strong passwords: Use strong passwords and change them regularly to keep accounts secure. 3. Antivirus and anti-malware software: Install and regularly update antivirus and Anti-malware software to ensure that the system is not infected with malware. 4. Timely backup: Back up important data regularly to prevent data loss and malware attacks. 5. Install a firewall: Install and configure a firewall to prevent unauthorized access. 6. Restrict access rights: Restrict user and administrator access rights to ensure that only authorized personnel can access systems and data. 7. Train employees: Conduct security training for employees to improve their security awareness and prevention capabilities. 8. Monitoring logs: Regularly monitor system logs to detect abnormal behaviors and attacks in a timely manner."]}

## E.4 MULTIPLE-CHOICE EXAMPLES IN OWL-BENCH

In Table 8, we give examples of different domains. The correct answers to the multiple-choice questions are marked in red, and each question has only one correct answer.

## F PROMPT TEMPLATES

### F.1 PROMPT FOR DATA QUALITY

The prompt we employ for GPT-4 is deliberately tailored to elicit insightful evaluations and is as follows:

> **Quality Prompt**: You are now a data grader. You will grade the data I provide according to my requirements, explain the reasons, and then give a piece of higher-quality data based on this piece of data.
> Please help me rate the following dialogue data in the field of operation and maintenance and explain the reasons. Require:
> 1. Scoring perspective: whether the problem belongs to the field of operation and maintenance; whether the problem description is clear; whether the answer is accurate; whether the problem has a certain meaning; whether the language is coherent; whether the problem is challenging and difficult.
> 2. Point scale: 5-point scale, 1 point: very poor; 2 points: slightly poor; 3 points: barely qualified; 4 points: usable; 5 points: excellent.
> 3. Please rate the problem and attach reasons. If the score is lower than 4 points, a higher quality data will be generated based on this piece of data.
> 4. Format: You can only return a parsable json format data, no other content. For example: "score": 4, "reason": "", "modified-data": "". Among them, score represents the score for this question, reason represents the reason for the score, and states the advantages and disadvantages of the data, and modified-data represents You generated a new, higher-quality data based on the above data. Compared with the data provided, this new data solves the shortcomings you mentioned above and is directly available.
> 5. All reasons are written in reason.
> 6. If the score is lower than 4 points, modified-data must be provided.
> 7. Modified-data contains a complete piece of data that is directly available, and the quality must be higher and more in line with the quality of ChatGPT's training data. If null needs to be output, replace it with None. Now please follow the above requirements to annotate the following conversation data and return your annotated results in pure json form: "".

| Domain | Infrastructure operation and maintenance |
|---|---|

Suppose you are an operation and maintenance engineer of a large network company, and you need to write a Bash script to regularly monitor and record the CPU and memory usage of the server. Which of the following commands can be used in a Bash script to gather this information?
(A) ifconfig (B) Use automated VM-based recovery strategies
(C) Use a multi-region or multi-AZ deployment (D) Perform regular physical maintenance on the server

| Domain | Middleware operation and maintenance |
|---|---|

In the disaster recovery and backup solution of web applications based on cloud technology, which of the following methods can effectively reduce the single point of failure rate of the system and make the program show high availability?
(A) Back up all data and applications in only a single region (B) Use automated VM-based recovery strategies
(C) Use a multi-region or multi-AZ deployment (D) Perform regular physical maintenance on the server

| Domain | Application business operation and maintenance |
|---|---|

You are operating a large-scale e-commerce website, which will face huge traffic pressure during special festivals such as "Black Friday" or "Double Eleven". Which of the following is a strategy you might use to cope with this stress?
(A) All website updates and maintenance operations are suspended during the festive period.
(B) Increase the number of servers ahead of time to handle expected traffic growth.
(C) Partition the database to enhance query efficiency and concurrent processing capabilities.
(D) Limit the number of purchases per user during the holiday season.

| Domain | System architecture operation and maintenance |
|---|---|

When designing and implementing a high-availability, scalable cluster system, which of the following is not an implementation solution that needs to be considered?
(A) Data storage: adopt a distributed storage solution to store data in multiple nodes, such as using NoSQL database or distributed file system and other technologies.
(B) Load balancing: use hardware load balancer or software load balancer to realize the distribution of requests to ensure the load balance of each node.
(C) Fault tolerance: through technologies such as backup, redundancy, and failover, it is guaranteed that the system can still maintain availability in the event of node failure or network failure.
(D) Data compression: By compressing system data, storage space requirements are reduced and storage efficiency is improved.

| Domain | Database operation and maintenance |
|---|---|

In a system disaster recovery scenario, which of the following options is not primarily used to reduce data loss?
(A) Using the master-backup architecture, that is, all the data and configuration of the master server are copied to the backup server in real time.
(B) Regularly back up critical data and configuration files, with full backups available daily and incremental backups hourly.
(C) For database servers, use database replication techniques, such as master-slave replication or multi-master replication, to replicate data to backup servers in real time.
(D) Establish an automated disaster recovery process, including steps such as system recovery, data recovery, and configuration recovery.

| Domain | Network operation and maintenance |
|---|---|

A DNS server architecture includes components such as master-slave DNS servers, DNS cache servers, and domain name resolution routers. Which of these components is mainly responsible for distributing DNS query traffic of users to different DNS servers, so as to improve the reliability and scalability of the network?
(A) primary DNS server (B) from DNS server (C) DNS cache server (D) domain name resolution router

| Domain | Software architecture operation and maintenance |
|---|---|

The software development process includes sub-processes such as requirements analysis, general design, detailed design, coding, testing, and maintenance. The overall structure design of the software is completed in the ( ) sub-process.
(A) demand analysis (B) outline design (C) detailed design (D) Write code

| Domain | Operating system operation and maintenance |
|---|---|

Which command can be used to check the file system under Linux?
(A) ls (B) dd (C) grep (D) fsck

| Domain | Information security operation and maintenance |
|---|---|

From the following logs, what conclusions can be drawn? [2022-01-01 10:00:00] INFO: User login successful, username: admin [2022-01-01 10:05:00] WARNING: Memory usage exceeded threshold, performing memory cleanup [2022-01-01 10 :10:00] ERROR: File system corruption, performing repair [2022-01-01 10:15:00] DEBUG: Network connection problem detected, investigating
(A) The system has a memory leak (B) Login failed for user admin
(C) The file system was repaired successfully (D) The system is processing a network connection problem

Table 8: The prompt and response in different samples in Owl-Bench.

**A Case for Data Quality**: Raw data: { "instruction": "How to install mysql?", "input": "", "output": "1. Download MySQL Community Server; 2. Install MySQL Community Server; 3. Start MySQL Community Server; 4. Set the MySQL root account password." }
Score: { "score": 2, "reason": "The question does not entirely belong to the field of operation and maintenance, and the description of the problem is not clear enough, and it does not explain in what environment mysql is installed. In addition, the answer is not satisfactory enough and is relatively simple. Although the operation steps are given, there is no explanation Users would like to get more extensive answers regarding the specific operation methods of each step, such as explaining how to set the MySQL root account password next.", "modified-data": { "instruction": "How to install mysql on mac? Please explain the specific installation steps in detail", "input": "", "output": "There are many ways to install MySQL on Mac. Here is one of the common methods: 1. Download MySQL Community Server: Download MySQL Community Server for Mac OS X from the official MySQL website at https://dev.mysql.com/downloads/mysql/. 2. Install MySQL Community Server: Double-click the downloaded dmg file, and then follow the prompts to install. 3. Start MySQL Community Server: Open the "MySQL" option in "System Preferences" and start MySQL Community Server. 4. Set the MySQL root account password: Enter the following command in the terminal to set the root account password: "' /usr/local/mysql/bin/mysqladmin -u root password 'yourpassword' "' (Note: Replace 'yourpassword' with your own password) 5. Test MySQL: Enter the following command in the terminal to test whether MySQL has been successfully installed and running: "' /usr/local/mysql/bin/mysql -u root -p "' Then enter the root account password. If you successfully log in to MySQL, the installation and configuration have been completed. The above is a common way to install MySQL in Mac, but there are other methods and you can choose according to your needs and preferences. " } }

## F.2 PROMPT FOR SINGLE-SCORE MODE

**Prompt for single-score mode:**
You are a helpful assistant.
[Instruction] Please act as an impartial judge and evaluate the quality of the response provided by an AI assistant to the user question displayed below. Your evaluation should consider correctness and helpfulness. You will be given a reference answer and the assistant's answer. Begin your evaluation by comparing the assistant's answer with the reference answer. Identify and correct any mistakes. Be as objective as possible. After providing your explanation, you must rate the response on a scale of 1 to 10 by strictly following this format: "[[rating]]", for example: "Rating: [[5]]".
[Question]: ...
[The Start of Reference Answer] .... [The End of Reference Answer]
[The Start of Assistant's Answer] .... [The End of Assistant's Answer]

## F.3 PROMPT FOR PAIRWISE-SCORE MODE

**Prompt for pairwise-score mode:**
Please act as an impartial judge and evaluate the quality of the responses provided by two AI assistants to the user question displayed below. Your evaluation should consider correctness and helpfulness. You will be given a reference answer, assistant A's answer, and assistant B's answer. Your job is to evaluate which assistant's answer is better. Begin your evaluation by comparing both assistants' answers with the reference answer. Identify and correct any mistakes. Avoid any positional biases and ensure that the order in which the responses were presented does not influence your decision. Do not allow the length of the responses to influence your evaluation. Do not favor certain names of the assistants. Be as objective as possible. After providing your explanation, output your final verdict by strictly following this format: "[[A]]" if assistant A is better, "[[B]]" if assistant B is better, and "[[C]]" for a tie. [User Question]:...
[The Start of Reference Answer] ... [The End of Reference Answer]
[The Start of Assistant A's Answer] ... [The End of Assistant A's Answer]
[The Start of Assistant B's Answer] ... [The End of Assistant B's Answer]

## F.4 PROMPT FOR LOG PARSING

**Log Parsing Prompt**: Convert the following log into a standardized template by identifying and replacing the variable parts with a * and retain the keywords: [Input Log]

### F.5 Prompt for Anomaly Detection

**Anomaly Detection prompt**: Classify the given logs into normal and abnormal categories. Do it with these steps: (a) Mark it normal when values (such as memory address, floating number and register value) in a log are invalid. (b) Mark it normal when lack of information. (c) Never consider $\langle * \rangle$ and missing values as abnormal patterns. (d) Mark it abnormal when and only when the alert is explicitly expressed in textual content (such as keywords like error or interrupt).

## G  Mathematical Derivation

### G.1  Derivation of NBCE

Assuming $T$ is the target sentence to be generated and $S = (S_1, \ldots, S_n)$ are given sets of relatively independent context separated from the original sentence $S$, we need to generate the target sentence $T$ based on the independent sequence $S = (S_1, \ldots, S_n)$. The target sentence can be estimated as $P(T|S_1, \ldots, S_n)$:

$$P(T|S) = \frac{P(S|T)P(T)}{P(S)} \approx \prod_{i=1}^{N} P(S_k|T)P(T) \approx \frac{1}{P(T)^{n-1}} \prod_{k=1}^{N} P(S_k|T) \tag{13}$$

where $\frac{P(S|T)P(T)}{P(S)}$ follows the independent assumption and the item $P(S)$ is omitted. Equation 13 can be further rewritten as with $P(S_k|T) = \frac{P(T|S_k)}{P(T)}$

The log-likelihood of $P(T|S)$ can be represented by:

$$\log P(T|S) = \sum_{k=1}^{n} \log P(S_k|T) - (n-1)\log P(T) \tag{14}$$

### G.2  Derivation of HMCE

Let $T$ be the text that needs to be generated. Let $S := (S_1, ..., S_n)$ be the previous contexts, and that their overall length has exceeded the training length. We want to generate $T$ based on $S_1, ..., S_n$, that is, we want to estimate $p(T|S)$.

We assume that $S$ follows a conditional homogeneous Markov chain structure, that is

$$p(S_i|S_{1:i-1}, T) = p(S_i|S_{i-1}, T)$$

Also, by the Bayes' formula, we have

$$p(T|S) \propto p(S|T)p(T)$$

And based on the conditional probability, we have

$$p(S, T) = p(S|T)p(T) = p(S_n|S_{1:n-1}, T)p(S_{1:n-1}, T)$$

Thus we reduce the formula

$$p(S|T) = p(S_n|S_{1:n-1}, T)p(S_{1:n-1}|T)$$
$$= ...$$
$$= \prod_{i=2}^{n} p(S_i|S_{i-1}, T)p(S_1|T)$$

And

$$p(S_i|S_{i-1}, T) \propto p(T|S_i, S_{i-1})/p(T|S_{i-1})$$

Hence

$$p(T|S) \propto \prod_{i=2}^{n} \frac{p(T|S_i, S_{i-1})}{p(T|S_{i-1})} p(T|S_1)$$

Therefore

$$\log p(T|S) \propto \sum_{i=2}^{n} p(T|S_i, S_{i-1}) - \sum_{i=2}^{n-1} p(T|S_i)$$

## H    DETAILS FOR DATA COLLECTION IN OWL-BENCH

Our primary data source comprises practice exams that have been made freely accessible on the Internet. These practice exams, often used for honing the skills of aspiring professionals, serve as a valuable repository of real-world questions and scenarios within the operations and maintenance domain. Additionally, to further enrich the quality and authenticity of our benchmark, we have collaborated with operations and maintenance experts. Approximately 200 questions meticulously vetted by these experts have been seamlessly integrated into the Owl-Bench. This collaborative effort not only bolsters the benchmark's credibility but also infuses it with domain-specific expertise. It is worth emphasizing our commitment to openness and knowledge sharing. All the questions within Owl-Bench are slated to be open-sourced, a testament to our dedication to fostering a collaborative and transparent environment within the research community. This initiative is poised to facilitate broader access to high-quality data for the advancement of operations and maintenance-related research and innovation.

## I    RELATED WORKS

**Language Models.**    Language is a distinct human skill that evolves throughout life and starts developing in early childhood (Bai et al., 2023a; Guo et al., 2023b; 2022; Liu et al., 2022). Machines lack an inherent capacity to naturally comprehend and employ human language without the aid of advanced artificial intelligence (AI) algorithms. Language modeling based on the self-supervised learning training objective and large-scale data has been widely used to acquire contextual representations. Pre-training a large Transformer encoder/decoder (Vaswani et al., 2017; Chi et al., 2020; Yang et al., 2023b) brings significant improvement for various downstream natural language processing tasks. Besides, pre-training a Transformer decoder (Brown et al., 2020b) is beneficial for unconditional text generation. Enlarging Pre-training Language Models (PLMs) by increasing their model or data size brings in huge performance improvement in various tasks, adhering to a known scaling principle. To explore this, numerous studies have pushed the boundaries by training increasingly larger PLMs (Chowdhery et al., 2022a; Anil et al., 2023; Touvron et al., 2023a;b; Xu et al., 2023a; Ghosal et al., 2023; Bai et al., 2023b), such as the 175-billion parameter GPT-3 and the 540-billion parameter PaLM. The scaling laws of large language model (Kaplan et al., 2020; Aghajanyan et al., 2023) can guide the training of the large language model.

Despite primarily focusing on scaling model size while retaining similar architectures and pre-training tasks, these expansive PLMs exhibit distinct behaviors compared to smaller ones (Brown et al., 2020b; OpenAI, 2023). These extensive models showcase unforeseen capabilities, often referred to as "emergent abilities", which enable them to excel in intricate tasks. An exemplary illustration of the LLM application is observed in ChatGPT, which adapts LLMs from the GPT series to engage in dialogues, demonstrating a remarkable conversational prowess with humans. Fine-tuned LLMs on numerous datasets show promising results (Chung et al., 2022; Wang et al., 2023a; Luo et al., 2023; Wei et al., 2022; Chen et al., 2022), where the prompts used for instruction tuning can be created by humans or by LLMs themselves, and follow-up instructions can be used to refine the generation. An approach (Wei et al., 2022; Chen et al., 2022) related to instruction tuning is chain-of-thought prompting, where models are prompted to explain their reasoning when given a complex problem, in order to increase the likelihood that their final answer is correct. RLHF (Ouyang et al., 2022) has emerged as a powerful strategy for fine-tuning Large Language Models, enabling significant improvements in their performance. In our work, we collect the Owl-Instruct to train and evaluate the proposed OWL.

**Specialized Large Language Models.**    The value of training specialized decoder-only large language models is widely used in many fields, such as Financial LLMs (Wu et al., 2023; Yang et al., 2023a), Code LLMs (Rozière et al., 2023; Luo et al., 2023), and Layer LLMs (Cui et al., 2023; Nguyen, 2023). Common strategies involve training specialized models to continue to pre-train an existing model using new domain-specific data. In the field of IT operations, natural language processing (NLP) technologies have assumed a paramount role (Zhang et al., 2017; Messaoudi et al., 2018; Fu et al., 2009; Du & Li, 2016; He et al., 2017; Zhang et al., 2019), such as log analysis and parsing. The large language model for IT Operations can handle question-answering tasks in

many fields, such as infrastructure operation&maintenance, middleware operation&maintenance, and software architecture operation&maintenance.

