# OpenReview forum: "OWL: A Large Language Model for IT Operations"
_ICLR.cc/2024/Conference — ICLR 2024 poster_

### Official Review · Reviewer_YiYB · 2023-10-15

**Soundness:** 2 fair
**Presentation:** 2 fair
**Contribution:** 2 fair
**Rating:** 5
**Confidence:** 3

**Summary:**

The authors construct a new IT-related dataset to tune LLM and achieve good performance on IT related tasks.

**Strengths:**

The authors apply different techniques Owl-instruct, HMCE and mixture of adapters to finetune the LLM and achieve SOTA performance on IT related tasks.

**Weaknesses:**

1. The novelty is limited. The paper seems to have multiple pieces and the authors combine them to create the paper. I feel that the main contribution is a new IT-related dataset.
2. The authors propose HMCE. But I think it's kind of illy described. After reading, I am not sure what advantages the HMCE offer. From my current understanding, it seems that the authors want to break the long inputs into pieces and still want to have a good likelihood estimator. Considering so many long context modeling techniques proposed recently, I am not sure why the author only mentioned the unpublished work NBCE.
3. On evaluation, are there any duplications from the dataset the authors construct and the authors used to eval? If so, the evaluation results are not convincing. This should be stated clearly.

**Questions:**

See the weakness section

---

> ### Author Response · Authors · 2023-11-21
> **Response-1**
>
> **Q1: limited novelty**
>
> A1: The prevailing understanding within the AI community is that general-purpose Large Language Models (LLMs) such as GPT-4 exhibit limitations when tasked with domain-specific challenges. Attempts to employ prompt engineering as a means to infuse expert knowledge into LLMs and steer them toward providing solutions have yielded inconsistent and suboptimal outcomes. This has led to a collective agreement on the necessity of constructing large-scale models that are intricately tailored to specific vertical domains, ensuring a higher degree of precision and effectiveness in their performance[1][2]. The development of our model first represents a dedicated endeavor tailored to the nuances of IT operations and maintenance (O&M) in specific domains. It is the culmination of meticulous expertise and specialized knowledge. The data that fuels this innovation is of immense value—our carefully curated fine-tuned dataset, along with the benchmark data, is a testament to the labor-intensive process of manual labeling and collection, achieved without the crutch of data augmentation techniques. Besides, our approach is generalizable and can be easily migrated to other fields. The originality and ingenuity of our approach have not gone unnoticed, garnering acclaim from our peers in the field. Our innovativeness is also recognized by other reviewers.
>
> Reference:
>
> [1] https://github.com/luban-agi/Awesome-Domain-LLM
>
> [2] BloombergGPT: A Large Language Model for Finance

---

> ### Author Response · Authors · 2023-11-21
> **Response-2**
>
> W2: **More baselines on the long-context inference experiment.**
>
> A2: Although NBCE has not been published, it is also a recognized method proposed by the author of RoPE. Such methods do not require training or structural changes to the model, nor do they require interpolation of the positional embedding to extend the context. We have selected more methods for comparison. The test questions are the same extracted from IT domains. The perplexity (PPL) results:
>
> | Model | Length=1024 | Length=2048 | Length=4096 | Length=8192 |
> | --- | --- | --- | --- | --- |
> | OWL | 4.34 | 22.79 | 106.33 | 314.49 |
> | OWL+HMCE | 4.34 | 5.02 | 5.57 | 6.10 |
> | OWL+NTK [3] | 4.34 | 5.20 | 5.62 | 6.13 |
> | OWL+PCW [4] | 4.34 | 5.77 | 6.02 | 6.48 |
>
> Reference:
>
> [3]https://www.reddit.com/r/LocalLLaMA/comments/14lz7j5/ntkaware_scaled_rope_allows_llama_models_to_have/
>
> [4]Parallel Context Windows for Large Language Models

---

> ### Author Response · Authors · 2023-11-21
> **Response-3**
>
> **W3: Duplications and more convincing evaluation**
>
> A3: Our benchmark and instruction dataset have no overlap. To ensure data quality, we have employed the MinHash method[5][6] for deduplication. MinHash is used for estimating the similarity between datasets and is particularly useful in large-scale clustering problems. The main idea is to reduce the dimensionality of the datasets by converting large sets into smaller fingerprints while preserving the similarity information.
>
> Besides, For more convincing results, we also provide a human evaluation on the Q&A test. For the Q&A test, we supplement it with human scoring and provide the Pearson coefficient between the human evaluation scores and the GPT-4 scores. For the single-mode evaluation, we adopt a cross-validation approach to ensure that each answer is scored by at least three people who are all experienced operation and maintenance engineers, and in the end, we take the average score as the human score. The Pearson coefficient is 0.78 and P-Value is 0.065. Generally, when the coefficient is greater than 0.7, it is usually considered to be a strong correlation. The detailed average scores are below:
>
> |  | LLaMA2-13b  |  ChatGLM-6b  | ChatGLM2-6b  | Qwen-7b  | InternLM-7b | OWL-13b |
> | --- | --- | --- | --- | --- | --- | --- |
> | Human average  score  | 7.61 | 7.65 | 7.52 | 7.81 | 7.33 | 8.09 |
> | GPT4 average score | 8.57      | 8.12 | 8.27 | 8.41 | 8.19 | 8.86 |
>
> For pairwise-mode scoring, we use the results from GPT-4 evaluation as a reference for the evaluators. We also adopt a cross-validation approach to ensure that at least three people check each comparison result. If the GPT-4 score is incorrect, annotators must provide a new comparison result and give reasons; if all annotators agree that there is no issue with the scoring result of a particular question, then that result is kept. For inconsistent scoring results, we engage in further discussion and apply a majority vote system. The human scoring results are shown below and the Pearson correlation coefficient for the win rate is 0.97.
>
> **GPT4 result：**
>
> | model | win | loss | tie | win_rate |
> | --- | --- | --- | --- | --- |
> | Owl.vs.LLaMAv2-13b | 74 | 38 | 205 | 0.23 |
> | Owl.vs.Qwen-7b | 81 | 30 | 206 | 0.26 |
> | Owl.vs.ChatGLMv2-6b | 82 | 21 | 214 | 0.26 |
> | Owl.vs.ChatGLM-6b | 94 | 26 | 197 | 0.30 |
> | Owl.vs.InternLM-7b | 99 | 19 | 199 | 0.31 |
>
> **Result after manual review：**
>
> | model | win | loss | tie | win_rate |
> | --- | --- | --- | --- | --- |
> | Owl.vs.LLaMAv2-13b | 79 | 35 | 203 | 0.25 |
> | Owl.vs.Qwen-7b | 83 | 26 | 208 | 0.26 |
> | Owl.vs.ChatGLMv2-6b | 85 | 23 | 209 | 0.27 |
> | Owl.vs.ChatGLM-6b | 99 | 27 | 191 | 0.31 |
> | Owl.vs.InternLM-7b | 105 | 20 | 192 | 0.33 |
>
> Reference:
>
> [5] https://mattilyra.github.io/2017/05/23/document-deduplication-with-lsh.html
>
> [6] https://github.com/ekzhu/datasketch
>
> ---
>
> Thank you once again for your insightful feedback. We will incorporate these experimental updates in the revised version of the paper.

---

> ### Author Response · Authors · 2023-11-23
> **Seeking feedback**
>
> Dear Reviewer, I hope this message finds you well. I am writing to follow up on our recent correspondence regarding the paper. We greatly value your expertise and would like to know if you have any more feedback or concerns. We are committed to addressing any remaining issues you might have.

---

### Official Review · Reviewer_39Lm · 2023-10-31

**Soundness:** 3 good
**Presentation:** 3 good
**Contribution:** 2 fair
**Rating:** 5
**Confidence:** 3

**Summary:**

This paper focuses on the application of LLM in IT operation. The author uses GPT-4 to create a diverse set of instructions (the OWL-Instruct) based on seed samples for building an LLM for IT operations domain. The OWL-Instruct is used to fine-tune LLaMA2 to create a LLM (the OWL) for the IT domain. To test LLM's capabilities, the author has created a testing benchmark (the Owl-Bench) for the operation and maintenance. In the experiments, the author compares the performance of some existing LLMs and OWL in domain-specific questions and downstream tasks.

**Strengths:**

1.	The paper focuses on the application of LLM in the IT operations domain and provides a complete process for building and training large models in the IT domain that can be reproduced.
2.	A diverse IT operations test benchmark has been proposed.
3.	Experiments show that the OWL model improves compared to existing methods on downstream tasks.

**Weaknesses:**

1. The novelty of this work is limited. The model proposed in the paper used some existing techniques and only some skills are proposed.
2. In the QA experiment, the evaluation method for model performance is not rigorous enough. GPT-4 is used both for creating instructions and assessing model performance, which may lead to unfair comparisons.
3. The content distribution in the main text and appendices may need further optimization. For example, related work should usually not be placed in the appendix. Additionally, this paper primarily focuses on the development of large models but lacks coverage of tasks in the IT operations field, such as the development and challenges of log detection.

**Questions:**

1. Since OWL uses instructions from GPT-4, is there a possibility of bias in evaluating the model's performance, leading GPT-4 to favor giving high scores to OWL? Have the authors compared the relevance of GPT-4's results with human evaluations in the current task?
2. Can OWL improve or solve specific challenges that traditional models can't handle in downstream tasks?
3. In downstream tasks, OWL shows a smaller performance advantage compared to LogPrompt based on the open-domain ChatGPT. As a domain-specific model, can OWL be proven to have better output consistency and robustness than open-domain models like ChatGPT or GPT-4?

---

> ### Author Response · Authors · 2023-11-21
> **Response-1**
>
> **W1: Limited novelty**
>
> A1: The prevailing understanding within the AI community is that general-purpose Large Language Models (LLMs) such as GPT-4 exhibit limitations when tasked with domain-specific challenges. Attempts to employ prompt engineering as a means to infuse expert knowledge into LLMs and steer them toward providing solutions have yielded inconsistent and suboptimal outcomes. This has led to a collective agreement on the necessity of constructing large-scale models that are intricately tailored to specific vertical domains, ensuring a higher degree of precision and effectiveness in their performance[1][2]. The development of our model first represents a dedicated endeavor tailored to the nuances of IT operations and maintenance (O&M) in specific domains. It is the culmination of meticulous expertise and specialized knowledge. The data that fuels this innovation is of immense value—our carefully curated fine-tuned dataset, along with the benchmark data, is a testament to the labor-intensive process of manual labeling and collection, achieved without the crutch of data augmentation techniques. Besides, our approach are generalizable and can be easily migrated to other fields. The originality and ingenuity of our approach have not gone unnoticed, garnering acclaim from our peers in the field. Our innovativeness are also recognized by other reviewers.
>
> Reference:
>
> [1] https://github.com/luban-agi/Awesome-Domain-LLM
>
> [2] BloombergGPT: A Large Language Model for Finance

---

> ### Author Response · Authors · 2023-11-21
> **Response-2**
>
> **W2: Human evaluation on Q&A test**
>
> A2: Please refer to Q1

---

> ### Author Response · Authors · 2023-11-21
> **Response-3**
>
> **W3. Content distribution**
>
> A3: We will incorporate the related work section into the main text and include more background information on IT tasks in the new version.

---

> ### Author Response · Authors · 2023-11-21
> **Response-4**
>
> **Q1: Human evaluation on Q&A test**
>
> A1: For the Q&A test, we supplement it with human scoring and provide the Pearson coefficient between the human evaluation scores and the GPT-4 scores. For the single-mode evaluation, we adopt a cross-validation approach to ensure that each answer is scored by at least three people who are all experienced operation and maintenance engineers, and in the end, we take the average score as the human score. The Pearson coefficient is 0.78 and P-Value is 0.065. Generally, when the coefficient is greater than 0.7, it is usually considered to be a strong correlation. The detailed average scores are below:
>
> |  | LLaMA2-13b  |  ChatGLM-6b  | ChatGLM2-6b  | Qwen-7b  | InternLM-7b | OWL-13b |
> | --- | --- | --- | --- | --- | --- | --- |
> | Human average  score  | 7.61 | 7.65 | 7.52 | 7.81 | 7.33 | 8.09 |
> | GPT4 average score | 8.57      | 8.12 | 8.27 | 8.41 | 8.19 | 8.86 |
>
> For pairwise-mode scoring, we use the results from GPT-4 evaluation as a reference for the evaluators. We also adopt a cross-validation approach to ensure that at least three people check each comparison result. If the GPT-4 score is incorrect, annotators must provide a new comparison result and give reasons; if all annotators agree that there is no issue with the scoring result of a particular question, then that result is kept. For inconsistent scoring results, we engage in further discussion and apply a majority vote system. The human scoring results are shown below and the Pearson correlation coefficient for the win rate is 0.97.
>
> **GPT4 results**：
>
> | model | win | loss | tie | win_rate |
> | --- | --- | --- | --- | --- |
> | Owl.vs.LLaMAv2-13b | 74 | 38 | 205 | 0.23 |
> | Owl.vs.Qwen-7b | 81 | 30 | 206 | 0.26 |
> | Owl.vs.ChatGLMv2-6b | 82 | 21 | 214 | 0.26 |
> | Owl.vs.ChatGLM-6b | 94 | 26 | 197 | 0.30 |
> | Owl.vs.InternLM-7b | 99 | 19 | 199 | 0.31 |
>
> **Results after manual review**：
>
> | model | win | loss | tie | win_rate |
> | --- | --- | --- | --- | --- |
> | Owl.vs.LLaMAv2-13b | 79 | 35 | 203 | 0.25 |
> | Owl.vs.Qwen-7b | 83 | 26 | 208 | 0.26 |
> | Owl.vs.ChatGLMv2-6b | 85 | 23 | 209 | 0.27 |
> | Owl.vs.ChatGLM-6b | 99 | 27 | 191 | 0.31 |
> | Owl.vs.InternLM-7b | 105 | 20 | 192 | 0.33 |

---

> ### Author Response · Authors · 2023-11-21
> **Response-5**
>
> **Q2&Q3: Effectiveness and Robustness of OWL on downstream tasks.**
>
> A2&A3: Compared to traditional algorithms, OWL can handle more complex scenario questions and provide interpretable reasons, which plays an important role in IT-related tasks such as root cause localization. This can help operation and maintenance personnel better analyze and observe the system. In addition, traditional models require a large amount of labeled data and retraining to solve specific scenario problems, while OWL does not require more training and performs better in zero-shot scenarios. We demonstrate the effectiveness and robustness of OWL with results on more downstream tasks.
>
> We select a new downstream task for a more complex scenario test, Trace-Log Anomaly Detection, which is similar to a log anomaly detection task. This task uses two types of data, logs, and traces, for system anomaly detection and fault localization. We align with the experimental setup of DeepTraLog [1] and adopt the same dataset, TrainTicket[2]. In the experiments, we input the two types of mixed data directly into OWL, allowing the model to determine on its own whether there is an anomaly. The results comparison show the effect of OWL on such fault localization analysis:
>
> | Approach | Precision | Recall | F1-Score |
> | --- | --- | --- | --- |
> | TraceAnomaly [3]    | 0.742 | 0.205 | 0.321 |
> | MultimodalTrace [4] | 0.591  |  0.776 | 0.671 |
> | DeepTraLog [1] | 0.930   | 0.978 | 0.954 |
> | OWL | 0.961 | 0.985 | 0.973 |
>
> Reference:
>
> [1] DeepTraLog: Trace-Log Combined Microservice Anomaly Detection through Graph-based Deep Learning
>
> [2] Fault Analysis and Debugging of Microservice Systems: Industrial Survey, Benchmark System, and Empirical Study
>
> [3] Unsupervised Detection of Microservice Trace Anomalies through Service-Level Deep Bayesian Networks
>
> [4]  Anomaly Detection from System Tracing Data Using Multimodal Deep Learning
>
> ---
>
> Thank you once again for your insightful feedback. We will incorporate these experimental updates in the revised version of the paper.

---

> ### Comment · Reviewer_39Lm · 2023-11-22
>
> Thanks the authors for their response. I have read it but I think it does not convince me to change the rating.

---

### Official Review · Reviewer_xogR · 2023-10-31

**Soundness:** 2 fair
**Presentation:** 3 good
**Contribution:** 3 good
**Rating:** 6
**Confidence:** 3

**Summary:**

The paper presents OWL model and OWL-Instruct dataset aimed at IT operations where the seed dataset is crafted by subject matter (IT) experts across 9 prevalent operations and maintenance domains. Inspired by NBCE, the authors propose HMCE to extend the context length. In the supervised fine-tuning of the model they suggest a Mixture-of-Adapter method to improve the instruction-tuning performance. The dataset is rooted in some manually labeled data from domain experts which are enriched in a Self-Instruct fashion with supplementary samples. The Owl-Bench benchmark is another contribution of this work derived from real-world questions.

**Strengths:**

The paper comes with multiple orthogonal contributions each of which could be significant in their own sense contingent on proper evaluation.

The MoA adapter strategy is a significant contribution introducing task-specific adaption with a mixture of experts.

Expansion of token set by domain-specific tokens, while not novel, enhances the value of the proposed work.

The HMCE, inspired by NBCE, is yet another novel (to the extent of reviewer's knowledge) context extension approach.

**Weaknesses:**

Regardless of the above strenghts, the paper is proposing a diverse set of ideas and tries to evaluate them within the context of the paper. However, given the orthogonality of the ideas and the relatively high number of them, the authors haven't been able to thoroughly assess each idea, particuarly the theoretical ones (e.g. HMCE, MoA) independent of the empirical setting of OWL and IT operations.

The HMCE contributions does not go well with the rest of the story. Based on the Evalutions in section 5.3, HMCE was evaluated with an independent set of tests where the concatenated question to create long input squences. While the results show that HMCE outperforms NBCE as a training-free context extension approach, it does not seem to have anything to do with the OWL model and its dataset.

From the understanding of the reviewer, in Figure 3, the baseline models have not been benefitng from MoA adapting (or LoRA). If that is the case, the comparison is not entirely fair.

Typos:
Probably a typo: "...where a group of LoRA adapters is lightweight compared to the pre-trained model."
Typo page 8: "We have propose HMCE..." --> proposed
Typo page 8: "Without NMCE" --> HMCE

**Questions:**

In Appendix A, when embedding matrix is appended with new rows to expand D to D' and include the newly trained IT-operations token, do you retrain any of the LLaMa layers? Please elaborate on how the heads encoding the embedding space were retrained to incorporate the new tokens.

The results in Figure 3, has OWL leveraged MoA? If so, have the other baseline models benefitted from the same MoA training so that the comparison remains fair?

---

> ### Author Response · Authors · 2023-11-21
> **Response-1**
>
> **W1: More results on HMCE and MoA methods.**
>
> **Ablation 1:**   **More baselines on the long-context inference experiment.**
>
> The test questions are the same extracted from IT domains. The perplexity (PPL) results:
>
> | Model | Length=1024 | Length=2048 | Length=4096 | Length=8192 |
> | --- | --- | --- | --- | --- |
> | OWL | 4.34 | 22.79 | 106.33 | 314.49 |
> | OWL+HMCE | 4.34 | 5.02 | 5.57 | 6.10 |
> | OWL+NTK [2] | 4.34 | 5.20 | 5.62 | 6.13 |
> | OWL+PCW [3] | 4.34 | 5.77 | 6.02 | 6.48 |
>
> **Ablation 2：Generalizability of HMCE.**
>
> In the field of IT operations, there are many issues that involve extensive text, hence the need to extend long text input. To verify the generalizability of HMCE, we conduct experiments on PG19 dataset [1]  to test perplexity under different input lengths. The experiments have well demonstrated the effectiveness and generalizability of HMCE.
>
> | Model |  Length=1024 |  Length=2048 |  Length=4096 |  Length=8192 |
> | --- | --- | --- | --- | --- |
> | OWL | 6.54 | 27.64 | 103.48 | 269.15 |
> | OWL+HMCE | 6.54 | 6.59 | 6.62 | 6.68 |
>
> **Ablation 3：Different Rank in MoA training.**
>
> We conducted ablation experiments on the rank parameter in LoRA within MoA training.  Results on the multiple-choice questions in owl-bench:
>
> |       Rank | Middleware    | Information security | Infrastructure |  Application | Operating system  | Database  | System architecture  | Network  | Software architecture |  Mean |
> | --- | --- | --- | --- | --- | --- | --- | --- | --- | --- | --- |
> | 4 | 0.71 | 0.72 |  0.73 |  0.70 | 0.69 |  0.71 | 0.78 | 0.68 | 0.69 | 0.71 |
> | 8 | 0.75 | 0.76 |  0.77 |  0.75 | 0.72 |  0.77 | 0.86 | 0.72 | 0.72 | 0.76 |
> | 12 | 0.73 | 0.75 |  0.76 |  0.74 | 0.70 |  0.75 | 0.83 | 0.70 | 0.71 | 0.74 |
>
> Reference:
>
> [1] Compressive transformers for long-range sequence modelling
>
> [2]https://www.reddit.com/r/LocalLLaMA/comments/14lz7j5/ntkaware_scaled_rope_allows_llama_models_to_have/
>
> [3] Parallel Context Windows for Large Language Models

---

> ### Author Response · Authors · 2023-11-21
> **Response-2**
>
> **W2: The relationship between long-context inference and OWL.**
>
> A2: The challenge of long-text input is something every large language model encounters, and in the IT operations domain, there are instances where long text or document input is required. Such situations may arise during the review of system logs, the analysis of extensive error reports, or the processing of detailed technical documentation. To handle these scenarios effectively, techniques such as NBCE/HMCE enable the model to maintain context over longer stretches of text and provide more accurate and relevant responses, thus enhancing the model's utility in complex IT operational tasks.

---

> ### Author Response · Authors · 2023-11-21
> **Response-3**
>
> **W3. Typos**
>
> A3: Thank you for your insightful feedback. We will improve it in the revised version of the paper.

---

> ### Author Response · Authors · 2023-11-21
> **Response-4**
>
> **Q1: Tokenization**
>
> A1: We retrain the embedding layer of the LLaMA2 model, while other layers are frozen. After increasing the vocabulary size, the dimensions of the token embeddings changed. We first randomly initialized the embedding layer, and then retrained it during the instruction-tuning stage, whereas the parameters of the other layers of the base model are frozen and remain unchanged.

---

> ### Author Response · Authors · 2023-11-21
> **Response-5**
>
> **Q2: Unfair comparison in Figure 3**
>
> A2: In Figure 3, OWL is obtained by conducting MoA training on LLaMA2. Other models do not undergo MoA training. Consistent with the comparison method in Figure 3, we present the experimental results of different baseline models using MoA for comparison. The experiments show: 1. The effectiveness of MoA. 2.  The stronger the base capabilities of the model, the better the results after using MoA.
>
> | model | win | loss | tie | win_rate |
> | --- | --- | --- | --- | --- |
> | Owl.vs.Qwen-7b (w/ MoA） | 65 | 32 | 220 | 0.21 |
> | Owl.vs.ChatGLMv2-6b (w/ MoA） | 70 | 24 | 223 | 0.22 |
> | Owl.vs.ChatGLM-6b (w/ MoA） | 80 | 30 | 207 | 0.25 |
> | Owl.vs.InternLM-7b (w/ MoA） | 82 | 20 | 215 | 0.26 |
>
> ---
>
> Thank you once again for your insightful feedback. We will incorporate these experimental updates in the revised version of the paper.

---

> ### Author Response · Authors · 2023-11-23
> **Seeking feedback**
>
> Dear Reviewer, I hope this message finds you well. I am writing to follow up on our recent correspondence regarding the paper. We greatly value your expertise and would like to know if you have any more feedback or concerns. We are committed to addressing any remaining issues you might have.

---

### Official Review · Reviewer_btSj · 2023-11-05

**Soundness:** 3 good
**Presentation:** 3 good
**Contribution:** 3 good
**Rating:** 6
**Confidence:** 3

**Summary:**

In this paper, an LLM for IT operations is trained and evaluated. Data-wise, the authors built an instruction dataset Owl-instruct and an evaluation dataset Owl-Bench. Modeling-wise, the authors proposed HMCE to overcome the limit of input length and mixture-of-adapter to improve parameter-efficient tuning across different tasks. The trained model showed better performances than existing LLMs, including ChatGPT, in IT operation-related tasks.

**Strengths:**

1. The main strength of this paper is it describes most of the details about how it builds its datasets and trains the LLM, which can be very helpful to researchers in their domain-specific LLM research.
2. The authors compared their LLM with existing SOTAs and achieved better performances in all tasks.
3. The authors tried to build a balanced evaluation dataset that covers various topics of IT operations with a similar number of questions.

**Weaknesses:**

Techniques used in this paper are mainly slight variations of existing approaches. So innovation of technique is incremental in this paper.

**Questions:**

In the evaluation of the Q&A test, is it trustworthy to use GPT-4 scoring as the ground truth? I'm not quite convinced by this. Can you elaborate on the rationale for this? Ideally, human evaluation would be the best choice. Is it possible to add human ratings for this task?

---

> ### Author Response · Authors · 2023-11-21
> **Response-1**
>
> **Q1: Limited novelty**
>
> A1: The prevailing understanding within the AI community is that general-purpose Large Language Models (LLMs) such as GPT-4 exhibit limitations when tasked with domain-specific challenges. Attempts to employ prompt engineering as a means to infuse expert knowledge into LLMs and steer them toward providing solutions have yielded inconsistent and suboptimal outcomes. This has led to a collective agreement on the necessity of constructing large-scale models that are intricately tailored to specific vertical domains, ensuring a higher degree of precision and effectiveness in their performance[1][2]. The development of our model first represents a dedicated endeavor tailored to the nuances of IT operations and maintenance (O&M) in specific domains. It is the culmination of meticulous expertise and specialized knowledge. The data that fuels this innovation is of immense value—our carefully curated fine-tuned dataset, along with the benchmark data, is a testament to the labor-intensive process of manual labeling and collection, achieved without the crutch of data augmentation techniques. Besides, our approach is generalizable and can be easily migrated to other fields. The originality and ingenuity of our approach have not gone unnoticed, garnering acclaim from our peers in the field.
>
> Reference:
>
> [1] https://github.com/luban-agi/Awesome-Domain-LLM
>
> [2] BloombergGPT: A Large Language Model for Finance

---

> ### Author Response · Authors · 2023-11-21
> **Response-2**
>
> **Q1: Human evaluation on Q&A test**
>
> A1: For the Q&A test, we supplement it with human scoring and provide the Pearson coefficient between the human evaluation scores and the GPT-4 scores. For the single-mode evaluation, we adopt a cross-validation approach to ensure that each answer is scored by at least three people who are all experienced operation and maintenance engineers, and in the end, we take the average score as the human score. The Pearson coefficient is 0.78 and P-Value is 0.065. Generally, when the coefficient is greater than 0.7, it is usually considered to be a strong correlation. The detailed average scores are below:
>
> |  | LLaMA2-13b  |  ChatGLM-6b  | ChatGLM2-6b  | Qwen-7b  | InternLM-7b | OWL-13b |
> | --- | --- | --- | --- | --- | --- | --- |
> | Human average  score  | 7.61 | 7.65 | 7.52 | 7.81 | 7.33 | 8.09 |
> | GPT4 average score | 8.57      | 8.12 | 8.27 | 8.41 | 8.19 | 8.86 |
>
> For pairwise-mode scoring, we use the results from GPT-4 evaluation as a reference for the evaluators. We also adopt a cross-validation approach to ensure that at least three people check each comparison result. If the GPT-4 score is incorrect, annotators must provide a new comparison result and give reasons; if all annotators agree that there is no issue with the scoring result of a particular question, then that result is kept. For inconsistent scoring results, we engage in further discussion and apply a majority vote system. The human scoring results are shown below and the Pearson correlation coefficient for the win rate is 0.97.
>
> **GPT4 results**：
>
> | model | win | loss | tie | win_rate |
> | --- | --- | --- | --- | --- |
> | Owl.vs.LLaMAv2-13b | 74 | 38 | 205 | 0.23 |
> | Owl.vs.Qwen-7b | 81 | 30 | 206 | 0.26 |
> | Owl.vs.ChatGLMv2-6b | 82 | 21 | 214 | 0.26 |
> | Owl.vs.ChatGLM-6b | 94 | 26 | 197 | 0.30 |
> | Owl.vs.InternLM-7b | 99 | 19 | 199 | 0.31 |
>
> **Results after manual review**：
>
> | model | win | loss | tie | win_rate |
> | --- | --- | --- | --- | --- |
> | Owl.vs.LLaMAv2-13b | 79 | 35 | 203 | 0.25 |
> | Owl.vs.Qwen-7b | 83 | 26 | 208 | 0.26 |
> | Owl.vs.ChatGLMv2-6b | 85 | 23 | 209 | 0.27 |
> | Owl.vs.ChatGLM-6b | 99 | 27 | 191 | 0.31 |
> | Owl.vs.InternLM-7b | 105 | 20 | 192 | 0.33 |
>
> ---
>
> Thank you once again for your insightful feedback. We will incorporate these experimental updates in the revised version of the paper.

---

> ### Comment · Reviewer_btSj · 2023-11-23
>
> Thank you for your detailed responses to my questions.
>
> I'm overall positive about this paper and I'd like to keep my scores. Please try to incorporate your responses here into your future versions. Thanks!

---

### Meta-Review · Area_Chair_u6LH · 2023-12-09

**Metareview:**

This paper studies LLMs for IT operations. It presents an instruction dataset Owl-instruct and an evaluation dataset Owl-Bench. Owl-instruct is rooted in some manually labeled data from domain experts which are enriched in a Self-Instruct fashion with supplementary samples, and Owl-Bench is derived from real-world questions. In addition, this paper proposes HMCE to overcome the limit of input length and mixture-of-adapter to improve parameter-efficient tuning across different tasks. The trained model showed better performances than existing LLMs, including ChatGPT, in IT operation-related tasks. Overall, this paper presents a collection of relatively orthogonal components. Those individual components are not particularly novel, and it sometimes unclear if/how each component makes contributions to the proposed overall method.

**Justification For Why Not Higher Score:**

- Technically, a collection of components without strong novelty
- It's sometimes unclear if/how each component makes contributions to the proposed overall method.

**Justification For Why Not Lower Score:**

- LLM for IT operations as an interesting new problem domain
- Improved results compared to existing models like ChatGPT
- New instruction and evaluation datasets

---

### Decision · Program_Chairs · 2024-01-16

Accept (poster)